# Identifying Mislabeled Data using the Area Under the Margin Ranking

**Geoff Pleiss**[†]
Columbia University
gmp2162@columbia.edu

**Tianyi Zhang**[†]
Stanford University
tz58@stanford.edu

**Ethan Elenberg**
ASAPP
eelenberg@asapp.com

**Kilian Q. Weinberger**
ASAPP, Cornell University

## Abstract

Not all data in a typical training set help with generalization; some samples can be overly ambiguous or outrightly mislabeled. This paper introduces a new method to identify such samples and mitigate their impact when training neural networks. At the heart of our algorithm is the *Area Under the Margin* (AUM) statistic, which exploits differences in the training dynamics of clean and mislabeled samples. A simple procedure—adding an extra class populated with purposefully mislabeled *threshold samples*—learns a AUM upper bound that isolates mislabeled data. This approach consistently improves upon prior work on synthetic and real-world datasets. On the WebVision50 classification task our method removes 17% of training data, yielding a 1.6% (absolute) improvement in test error. On CIFAR100 removing 13% of the data leads to a 1.2% drop in error.

## 1   Introduction

As deep networks become increasingly powerful, the potential improvement of novel architectures in many applications is inherently limited by data quality. In many real-world settings, datasets may contain samples that are "weakly-labeled" through proxy variables or web scraping [e.g. 28, 35, 60]. Human annotators, especially on crowdsourced platforms, can also be prone to making labeling mistakes. Even the most celebrated and highly-curated datasets, like MNIST [31] and ImageNet [13], famously contain harmful examples. See Fig. 1 for suspicious examples detected by our proposed method—some are clearly mislabeled, others inherently ambiguous. Mislabeled training data are problematic for overparameterized deep networks, which can achieve zero training error even on randomly-assigned labels [65]. If a BIRD is mislabeled as a DOG, a model will learn overly specific filters—only applicable for this one image—which will result in overfitting and worse performance.

Our goal is to automatically identify and subsequently remove mislabeled samples from training datasets. Discarding these harmful data will reduce memorization and improve generalization. Perhaps more importantly, identifying mislabeled data allows practitioners to easily audit and curate their datasets. For example, a company might like to know about common labeling mistakes in order to reduce systematic error in their annotation pipeline. Large datasets may be too costly to manually inspect; therefore, an automated method should isolate mislabeled data with high precision and recall. Prior works have investigated multi-stage pipelines [e.g. 12, 20] or robust loss functions [e.g. 61, 67] for mislabeled sample identification. We instead wish to create a method that is fully "plug-and-play" with existing training methods for maximum compatibility and minimal implementation overhead.

---

[†]Work done while at ASAPP.

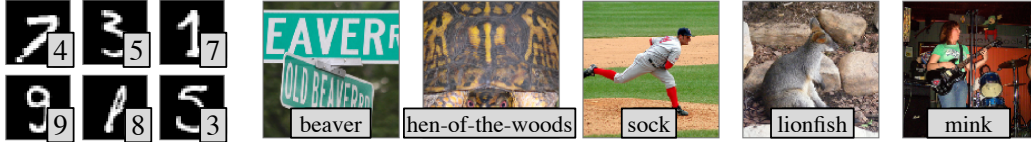

Figure 1: Images from **MNIST** (left) and **ImageNet** (right) with lowest *Area Under the Margin* (AUM) ranking (most likely to be mislabeled). AUMs are computed with LeNet/ResNet-50 models.

To this end, we propose a novel method that identifies mislabeled data *simply by observing a network's training dynamics*. Our method builds upon recent theoretical and empirical works [2, 5, 6, 33, 44] that suggest that dynamics of SGD contain salient signals about noisy data and generalization. Consider an image of a BIRD accidentally mislabeled as a DOG. Its memorization is the outcome of a delicate tension. During training, the gradient updates from the image itself encourage the network to (wrongly) predict the DOG label, whereas gradient updates from other training images encourage predicting BIRD through generalization. The opposing updates between the (incorrect) assigned label and the (hidden) true class membership are ultimately reflected in the logits during training.

To capture this phenomenon, we introduce the *Area Under the Margin* (AUM) statistic, which measures the average difference between the logit values for a sample's assigned class and its highest non-assigned class. Correctly-labeled data, which generalize from similarly-labeled examples, do not exhibit this tension and thus have a larger AUM than mislabeled data. To separate mislabeled samples from difficult but beneficial samples, we make a second contribution. We introduce an extra (artificial) class and purposefully assign a small percentage of *threshold* training data to this new class. All samples assigned to this new class are by definition mislabeled; therefore, we can use the AUM statistics of these points as a threshold to separate correctly-labeled data from mislabeled data.

The AUM statistic and threshold samples are trivially compatible with any classification network.[2] Implementing this method simply requires logging the model's logits during training. Training data whose AUM falls below the threshold can be confidently removed from the training set. On standard benchmark tasks, we improve upon the performance of existing methods simply by removing identified mislabeled samples. We are also able to clean many real-world datasets—including WebVision and Tiny ImageNet—for improved classification performance. Most surprisingly, *removing* 13% *of the CIFAR100 dataset results in a* 1.2% *reduction in test-error* for a ResNet-32 model.

## 2 Related Work

Learning with noisy data has been well studied [e.g. 16, 17, 42, 69]. Here we note several prior works, but please refer to [1] for a complete review. Within the context of deep learning, researchers have proposed novel training architectures [18], model-based curriculum learning schemes [26, 52], label correction [63], and robust loss functions [38, 40, 50, 61, 67]. Recent work has developed theoretical guarantees for certain forms of regularization [23, 34]. Our work aims not only to improve model robustness with minimal changes to the training procedure, but also to increase training set quality.

Our work shares a similar pipeline with Brodley and Friedl [11], where we first identify mislabeled examples before training a classifier on a cleaned dataset. In particular, we identify mislabeled samples through a ranking metric coupled with a learned threshold (as suggested by [42, 45]). There are many deep learning approaches that explicitly or implicitly identify mislabeled data. Some methods filter data through cross-validation [12], influence functions [58], or auxiliary networks [25]. Many approaches use signals from training as a proxy for label quality [19, 46, 54, 56, 62, 64, 65]. Arazo et al. [3] and Li et al. [33] fit the training losses of training data with two-component mixture models to separate clean from mislabeled data. We similarly use training dynamics; however, we rely on a metric (AUM) that is less prone to confusing difficult samples for mislabeled data. Moreover, we rely on threshold samples rather than a parametric mixture to separate mislabeled data. Some research examines a relaxed setting where a small set of data is assumed to be free of mislabeled examples [22, 32, 36, 51, 55]. This paper considers the more restrictive setting where no subset of the training data can be trusted.

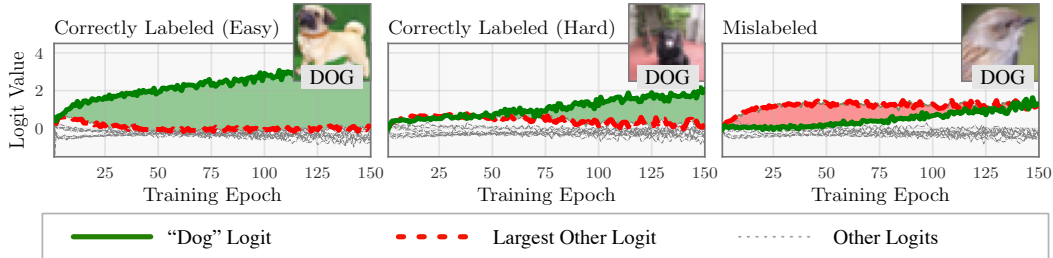

Figure 2: Illustration of the *Area Under the Margin* (AUM) metric. The graphs display logit trajectories for easy-to-learn dogs (left), hard-to-learn dogs (middle), and BIRDS mislabeled as DOGS (right). (Each plot's logits are averaged from 50 CIFAR10 training samples, $40\%$ label noise.) AUM is the shaded region between the DOG logit and the largest other logit. Green/red regions represent positive/negative AUM. Correctly-labeled samples have larger AUMs than mislabeled samples.

**Relation to semi-supervised learning/data augmentation.** There has been recent interest in combining noisy-dataset learning with semi-supervised learning and data augmentation techniques. These approaches identify a small set of correctly-labeled data and then use the remaining untrusted data in conjunction with semi-supervised learning [33, 39, 44, 47], pseudo-labeling [20, 68], or MixUp augmentation [3, 66]. Our paper is primarily concerned with how to identify correctly-labeled data rather than how to re-use mislabeled data. For simplicity, we discard data identified as mislabeled. However, any approach to re-integrate mislabeled data should be compatible with our method.

## 3   Identifying Mislabeled Data

We assume our training dataset $\mathcal{D}_{\text{train}} = \{\mathbf{x}_i, y_i\}_{i=1}^{N}$ consists of two data types. A **mislabeled sample** is one where the assigned label does not match the input. For example, $\mathbf{x}$ might be a picture of a BIRD and its assigned label $y$ might be DOG. A **correctly-labeled sample** has an assigned label that matches the ground-truth of the input. Some correctly-labeled examples might be "easy-to-learn" if they are common (e.g. $y = $ DOG, $\mathbf{x}$ is a golden retriever catching a frisbee). Others might be "hard-to-learn" if they are rare-occurrences (e.g. $y = $ DOG, $\mathbf{x}$ is an uncommon breed). In general, we assume both easy and hard correctly-labeled samples in $\mathcal{D}_{\text{train}}$ improve model generalization, whereas mislabeled examples hurt generalization. Our goal is to identify mislabeled data in $\mathcal{D}_{\text{train}}$ (i.e. samples that hurt generalization) simply by observing differences in training dynamics among samples.

**Motivation: measuring a sample's contribution to generalization with margins.** How do we determine whether or not a training sample contributes to/against generalization (and therefore is likely to be correctly-labeled/mislabeled)? The neural network community has proposed numerous metrics to quantify generalization—from parameter norms [e.g. 37] to noise stability [e.g. 4] to sharpness of minima [e.g. 29]. In this paper we utilize a metric based on the margin of training samples, which is a well-established notion for numerous machine learning algorithms [e.g. 7, 53, 57, 59]. Recent theoretical and empirical analyses suggest that margin distributions are predictive of neural network generalization [8, 15, 26, 43]. We extend this line of work by using the margin of the final layer to identify poorly generalizing training data and ultimately improve classifier performance. While other metrics have been investigated [e.g. 27], margins are advantageous because 1) they are simple and efficient to compute during training; and 2) they naturally factorize across samples, making it possible to estimate the contribution of each data point to generalization. Designing novel metrics which satisfy these criteria remains an interesting direction for future work. In concurrent work, Northcutt et al. [46] similarly investigate the margin for identifying mislabeled data.

**Area Under the Margin (AUM) Ranking.** Let $(\mathbf{x}, y) \in \mathcal{D}_{\text{train}}$ be a sample and let $\mathbf{z}^{(t)}(\mathbf{x}) \in \mathbb{R}^c$ be its logits vector (pre-softmax output) at epoch $t$ (logit $z_i^{(t)}(\mathbf{x})$ corresponds to class $i$). The **margin** at epoch $t$ captures how much larger the (potentially incorrect) assigned logit is than all other logits:

$$M^{(t)}(\mathbf{x}, y) = \overbrace{z_y^{(t)}(\mathbf{x})}^{\text{assigned logit}} - \overbrace{\max_{i \neq y} z_i^{(t)}(\mathbf{x})}^{\text{largest other logit}}. \tag{1}$$

A negative margin corresponds to an incorrect prediction, while a positive margin corresponds to a confident correct prediction. A sample will have a very negative margin if gradient updates from similar samples oppose the sample's (potentially incorrect) assigned label. We hypothesize that, at any given epoch during training, a mislabeled sample will *in expectation* have a smaller margin than a correctly-labeled sample. We capture this by averaging a sample's margin measured at each training epoch—a metric we refer to as **area under the margin** (AUM):

$$\text{AUM}(\mathbf{x}, y) = \frac{1}{T} \sum_{t=1}^{T} M^{(t)}(\mathbf{x}, y), \tag{2}$$

where $T$ is the total number of training epochs. This metric is illustrated by Fig. 2, which plots the logits for various CIFAR10 training samples over the course of training a ResNet-32. Each of the 10 lines represents a logit for a particular class. The left and middle graphs display correctly-labeled DOG examples—one that is "easy-to-learn" (low training loss) and one that is "hard-to-learn" (high training loss). For these two samples, the DOG logit grows larger than all other logits. The green shaded region measures the AUM, which is positive and especially large for the easy-to-learn example. Conversely, the right plot displays a mislabeled DOG training sample. For most of training the DOG logit is much smaller than the BIRD logit (red line)—the image's ground truth class—likely due to gradient updates from similar-looking correctly-labeled BIRDS. Consequentially, the mislabeled DOG has a very negative AUM, signified by the red area on the graph. This motivates AUM as a ranking: we expect mislabeled samples to have a smaller AUM values than correctly labeled samples (Fig. 3).

**Separating clean/mislabeled AUMs with threshold samples.** In order to identify mislabeled data, we must determine a threshold that separates clean and mislabeled samples. Note that this threshold is dataset dependent: Fig. 3 displays a violin plot of AUM values for CIFAR10/100 with $40\%$ label noise. CIFAR10 samples have AUMs between -4 and 2, and most negative AUMs correspond to mislabeled samples. The values on CIFAR100 tend to be more extreme (between -7 and 5), and up to $40\%$ of clean samples have a negative AUM. With access to a trusted validation set, a threshold can be learned through a hyperparameter sweep. Here, we propose a more computationally efficient strategy to learn a threshold without validation data. During training we insert fake data—which we refer to as **threshold samples**—that mimic the training dynamics of mislabeled data. Data with similar or worse AUMs than threshold samples can be assumed to be mislabeled.

We construct threshold samples in a simple way: *take a subset of training data and re-assign their label to a brand new class*—i.e. a class that doesn't really exist. In particular, assume that our training set has $N$ samples that belong to $c$ classes. We randomly select $N/(c+1)$ samples and re-assign their labels to $c+1$ (adding an additional neuron to the network's output layer for the fake $c+1$ class). Choosing $N/(c+1)$ threshold examples ensures the extra class is as likely as other classes on average. Since the network can only raise the assigned $c+1$ logit through memorization, we expect a small and likely negative margin for threshold samples, just as with mislabeled examples. Fig. 3 displays the AUMs of threshold samples (dashed gray lines), which are indeed smaller than correctly-labeled AUMs (blue histogram).

Using an extra class $c+1$ for threshold samples has subtle but important properties. Firstly, all threshold samples are *guaranteed* to mimic mislabeled data. (Since threshold samples are constructed from potentially-mislabeled training data, assigning a random label in $[1, c]$ could accidentally "correct" some mislabeled examples.) Moreover, the additional $c+1$ classification task does not interfere with the primary classifiers. In this sense, the network and AUM computations are minimally affected by the threshold samples.

As a simple heuristic, we identify data with a lower AUM than the 99th percentile threshold sample as mislabeled. (While the percentile value can be tuned through a hyperparameter sweep, we demonstrate in Appx. B that identification performance is robust to this

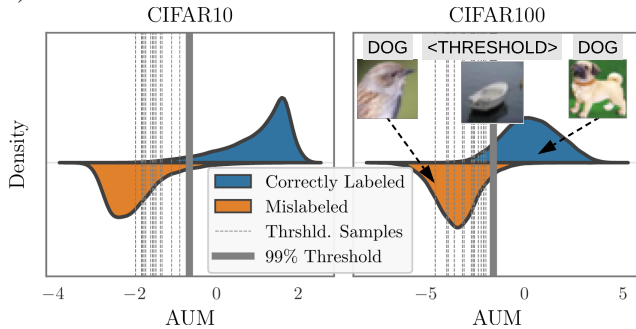

Figure 3: Illustrating the role of *threshold samples* on CIFAR10/100 with $40\%$ mislabeled samples. Histograms of AUMs for correctly-labeled (blue) and mislabeled samples (orange). Dashed lines represent the AUM values of threshold samples. The 99th percentile of threshold AUMs (solid gray line) separates correctly- and mislabeled data.

hyperparameter.) Fig. 3 demonstrates the efficacy of this strategy. The $99^{\text{th}}$ percentile threshold AUM (thick gray line) cleanly separates correctly- and mislabeled samples on noisy CIFAR10/100.

**Putting this all together,**   we propose the following procedure for identifying mislabeled data:

1. Create a subset $\mathcal{D}_{\text{THR}}$ of threshold samples:
2. Construct a modified training set $\mathcal{D}'_{\text{train}}$ that includes the threshold samples.

$$\mathcal{D}'_{\text{train}} = \{(\mathbf{x}, c+1) : \mathbf{x} \in \mathcal{D}_{\text{THR}}\} \cup (\mathcal{D}_{\text{train}} \backslash \mathcal{D}_{\text{THR}}).$$

3. Train a network on $\mathcal{D}'_{\text{train}}$ until the first learning rate drop, measuring the AUM of all data.
4. Compute $\alpha$: the $99^{\text{th}}$ percentile threshold sample AUM.
5. Identify mislabeled data using $\alpha$ as a threshold $\{(\mathbf{x}, y) \in (\mathcal{D}_{\text{train}} \backslash \mathcal{D}_{\text{THR}}) : \text{AUM}_{\mathbf{x},y} \leq \alpha\}$.

By stopping training before the first learning rate drop, we prevent the network from converging and therefore memorizing difficult/mislabeled examples. In practice, this procedure only allows us to determine which samples in $\mathcal{D}_{\text{train}} \backslash \mathcal{D}_{\text{THR}}$ are mislabeled. We therefore repeat this procedure using a different set of threshold samples to identify the remaining mislabeled samples. In total the procedure takes roughly the same amount of computation as training a normal network: two networks are trained up until the first learning rate drop (roughly halfway through the training of most networks).

## 4   Experiments

We test the efficacy of AUM and threshold samples in two ways. First, we directly measure the precision and recall of our identification procedure on synthetic noisy datasets. Second, we train models on noisy datasets after removing the identified data. We use test-error as a proxy for identification performance—removing mislabeled samples should improve accuracy, whereas removing correctly-labeled samples should hurt accuracy. In all experiments we do not assume the presence of any trusted data for training or validation. (See Appx. A for all experimental details.)

We note that our method and many baselines can be used in conjunction with semi-supervised learning [33, 44], pseudo-labeling [20, 68], or MixUp [3, 66] to improve noisy-training performance. Given that our focus is identifying mislabeled examples, we consider these to be complimentary orthogonal approaches. Therefore, we do not use these additional training procedures in any of our experiments.

### 4.1   Mislabeled Sample Identification

We use synthetically-mislabeled versions of **CIFAR10/100** [30], where subsets of $45,000$ images are used for training. We also consider **Tiny ImageNet**, a 200-class subset of ImageNet [13] with $95,000$ images resized to $64 \times 64$. We corrupt these datasets following a uniform noise model (mislabeled samples are given labels uniformly at random). We compare against several methods from the existing literature. Arazo et al. [3] fit the training losses with a mixture of two beta distributions (**DY-Bootstrap BMM**). Samples assigned to the high-loss distribution are considered mislabeled. The authors also propose using a mixture of two Gaussians (**DY-Bootstrap GMM**)—an approach also used by Li et al. [33]. **INCV** [12] iteratively filters training data through cross-validation. The remaining data are shared between two networks that inform each other about training samples with large loss (and are therefore likely mislabeled). We implement all methods on ResNet-32 models. For our method (**AUM**), as well as the BMM and GMM methods, we train networks for 150 epochs with no learning rate drops. We fit the BMM/GMMs to training losses from the last epoch. For INCV we use the publicly available implementation.

Fig. 4 displays the precision and recall of the identification methods at different noise levels. The most challenging settings for all methods are CIFAR100 and Tiny ImageNet with low noise. AUM tends to achieve the highest precision *and* recall in most noise settings, with precision and recall consistently $\geq 90\%$ in high-noise settings. It is worth emphasizing that the AUM model achieves this performance without any supervision or prior knowledge about the noise model.

### 4.2   Robust Training on Synthetic Noisy Datasets

To further evaluate AUM, we train ResNet-32 models on the noisy datasets after discarding the identified mislabeled samples. As a lower bound for test-error, we train a ResNet-32 following a

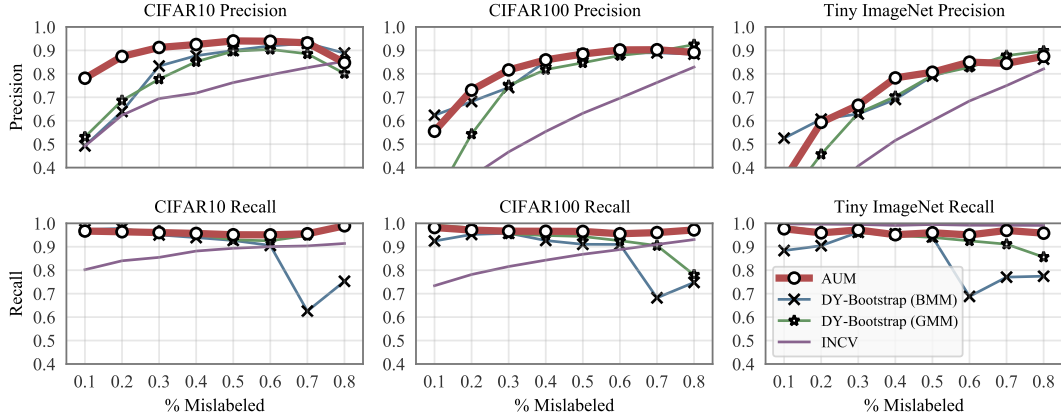

Figure 4: Precision/recall for identifying mislabeled data under uniform noise (ResNet-32 models).

Table 1: Test-error on CIFAR10/100 (ResNet-32) with synthetic mislabeled samples (uniform noise).

| Dataset | CIFAR10 | | | | CIFAR100 | | | |
|---|---|---|---|---|---|---|---|---|
| Noise | 0.20 | 0.40 | 0.60 | 0.80 | 0.20 | 0.40 | 0.60 | 0.80 |
| Standard | $25.0 \pm 0.3$ | $43.3 \pm 0.4$ | $63.3 \pm 0.4$ | $83.4 \pm 0.2$ | $50.4 \pm 0.2$ | $62.5 \pm 0.2$ | $76.2 \pm 0.4$ | $91.8 \pm 0.3$ |
| Random | $16.5 \pm 0.3$ | $22.8 \pm 0.8$ | $35.8 \pm 1.4$ | $55.6 \pm 2.8$ | $43.0 \pm 0.1$ | $51.2 \pm 0.2$ | $63.5 \pm 0.5$ | $83.7 \pm 0.9$ |
| Bootstrap [50] | $22.4 \pm 0.2$ | $37.4 \pm 0.4$ | $52.0 \pm 0.2$ | $68.8 \pm 0.5$ | $48.6 \pm 0.2$ | $58.9 \pm 0.2$ | $70.3 \pm 0.2$ | $89.8 \pm 0.8$ |
| MentorNet [25] | $13.3 \pm 0.1$ | $18.1 \pm 0.2$ | — | $66.0 \pm 2.5$ | $35.8 \pm 0.3$ | $42.5 \pm 0.2$ | — | $75.7 \pm 0.7$ |
| Co-Teaching [19] | $11.2 \pm 0.1$ | $13.5 \pm 0.1$ | $19.3 \pm 0.1$ | $80.7 \pm 0.6$ | $35.9 \pm 0.1$ | $39.8 \pm 0.2$ | $52.0 \pm 0.3$ | $89.1 \pm 0.7$ |
| D2L [40] | $12.3 \pm 0.2$ | $15.6 \pm 0.3$ | $27.3 \pm 0.6$ | Diverged | $46.0 \pm 1.0$ | $70.3 \pm 1.8$ | Diverged | Diverged |
| $L_{DMI}$ [61] | $14.1 \pm 0.1$ | $20.4 \pm 0.8$ | $34.9 \pm 1.2$ | $67.2 \pm 1.4$ | Diverged | Diverged | Diverged | Diverged |
| Data Param [52] | $17.9 \pm 0.2$ | $29.2 \pm 0.9$ | $50.7 \pm 0.7$ | $81.1 \pm 0.3$ | $43.7 \pm 1.4$ | $53.9 \pm 0.4$ | $67.2 \pm 2.3$ | $88.1 \pm 1.2$ |
| DY-Bootstrap [3] | $20.6 \pm 0.1$ | $31.2 \pm 1.4$ | $43.6 \pm 1.7$ | Diverged | $47.0 \pm 0.4$ | $57.0 \pm 0.3$ | $63.4 \pm 0.5$ | $87.2 \pm 0.5$ |
| INCV [12] | $10.5 \pm 0.1$ | $13.2 \pm 0.1$ | $18.9 \pm 0.3$ | $\mathbf{46.7 \pm 1.9}$ | $41.4 \pm 0.5$ | $44.6 \pm 0.2$ | $56.3 \pm 0.3$ | $76.3 \pm 0.6$ |
| AUM | $\mathbf{9.8 \pm 0.0}$ | $\mathbf{12.5 \pm 0.1}$ | $\mathbf{17.9 \pm 0.0}$ | $45.6 \pm 1.6$ | $\mathbf{34.5 \pm 0.2}$ | $\mathbf{38.7 \pm 0.1}$ | $\mathbf{47.0 \pm 0.5}$ | $\mathbf{68.3 \pm 0.7}$ |
| Oracle | $9.0 \pm 0.1$ | $9.7 \pm 0.4$ | $10.8 \pm 0.9$ | $12.6 \pm 1.7$ | $35.5 \pm 0.1$ | $39.0 \pm 0.2$ | $44.8 \pm 0.5$ | $55.1 \pm 0.4$ |

**Standard** training procedure on the full dataset. As an upper bound, we train an **Oracle** ResNet-32 on only the correctly-labeled data. We do not perform early stopping since we do not assume the presence of a clean validation set. These methods use the standard ResNet training procedure described in [21]. In addition, we compare against several baseline methods, including the **INCV** and **DY-Boostrap**[3] methods described above. **Bootstrap** [50] and **D2L** [40] interpolate the (potentially incorrect) training labels with the network's predicted labels. **MentorNet** [25] learns a weighting scheme with an LSTM.[4] **Co-teaching** identifies high-loss data with an auxiliary network; $L_{DMI}$ [61] uses a robust loss function; and **Data Parameters** [52] assigns a learnable weighting parameter to each sample/class. We also test a **Random Weighting** scheme [51], where all samples are assigned a weight from a rectified normal distribution (re-drawn at every epoch). We run all baseline experiments with ResNet-32 models, using publicly available implementations from the methods' authors.

Table 2: Test-error on Tiny ImageNet (ResNet-32) with synthetic mislabeled samples (uniform noise).

| Dataset | TinyImagenet | | | |
|---|---|---|---|---|
| Noise | 0.20 | 0.40 | 0.60 | 0.80 |
| Standard | $58.6 \pm 0.1$ | $66.3 \pm 0.2$ | $77.6 \pm 0.1$ | $92.4 \pm 0.1$ |
| Random | $54.3 \pm 0.1$ | $60.8 \pm 0.1$ | $70.7 \pm 0.2$ | $87.1 \pm 1.0$ |
| Data Param [52] | $54.2 \pm 0.1$ | $68.4 \pm 0.1$ | $84.3 \pm 0.3$ | $96.5 \pm 0.1$ |
| DY-Bootstrap [3] | $58.2 \pm 0.1$ | $63.7 \pm 0.2$ | $76.8 \pm 0.0$ | $93.2 \pm 0.4$ |
| INCV [12] | $54.8 \pm 0.1$ | $57.4 \pm 0.1$ | $63.8 \pm 0.2$ | $82.6 \pm 0.7$ |
| AUM | $\mathbf{51.1 \pm 0.2}$ | $\mathbf{55.3 \pm 0.1}$ | $\mathbf{62.7 \pm 0.3}$ | $\mathbf{79.2 \pm 0.2}$ |
| Oracle | $52.7 \pm 0.3$ | $56.5 \pm 0.2$ | $63.0 \pm 0.3$ | $71.2 \pm 0.1$ |

Table 1 displays the test-error of these methods on corrupted versions of CIFAR10/100. We observe several trends. First, there is a large discrepancy between the Oracle and Standard models (up to 62%). While most methods reduce this gap significantly, our identification scheme (AUM) achieves the lowest error in all settings. AUM recovers oracle performance on 20%/40%-noisy CIFAR10, and *surpasses* oracle performance on 20%/40%-noisy CIFAR100—simply by removing data from the training set. We hypothesize that AUM identifies mislabeled/ambiguous samples

Table 3: Test-error on real-world datasets (ResNet-32 for CIFAR/Tiny I.N., ResNet-50 for others).

| | | WebVision50 | Clothing100K | CIFAR10 | CIFAR100 | Tiny ImageNet | ImageNet |
|---|---|---|---|---|---|---|---|
| Standard | Error | 21.4 | 35.8 | $8.1 \pm 0.1$ | $33.0 \pm 0.3$ | $49.3 \pm 0.1$ | 24.2 |
| Data Param [52] | Error | 21.5 | 35.5 | $8.1 \pm 0.0$ | $36.4 \pm 1.4$ | $\mathbf{48.4 \pm 0.2}$ | $\mathbf{24.1}$ |
| DY-Bootstrap [3] | Error | 25.8 | 38.4 | $10.0 \pm 0.0$ | $34.9 \pm 0.1$ | $51.6 \pm 0.0$ | 32.0 |
| | (% Removed) | (4.6) | (12.1) | $(15.5 \pm 0.2)$ | $(7.3 \pm 0.1)$ | $(12.5 \pm 0.0)$ | (10.7) |
| INCV [12] | Error | 22.1 | $\mathbf{33.3}$ | $9.1 \pm 0.0$ | $38.2 \pm 0.1$ | $56.1 \pm 0.1$ | 29.5 |
| | (% Removed) | (26.2) | (25.2) | $(8.5 \pm 0.1)$ | $(27.4 \pm 0.1)$ | $(27.6 \pm 1.4)$ | (7.4) |
| AUM | Error | $\mathbf{19.8}$ | 33.5 | $\mathbf{7.9 \pm 0.0}$ | $\mathbf{31.8 \pm 0.1}$ | $48.6 \pm 0.1$ | 24.4 |
| | (% Removed) | (17.8) | (16.7) | $(3.0 \pm 0.1)$ | $(13.0 \pm 0.9)$ | $(19.9 \pm 0.1)$ | (2.7) |

in the standard (uncorrupted) training set. On Tiny ImageNet (Table 2), we compare AUM against Data Parameters, DY-Bootstrap, and INCV (three of the most recent methods that use identification). As with CIFAR10/100, AUM matches (or surpasses) oracle performance in most noise settings.

## 4.3 Real-World Datasets

**"Weakly-labeled" datasets.** We test the performance of our method on two datasets where the label-noise is unknown. WebVision [35] contains 2 million images scraped from Flickr and Google Image Search. It contains no human annotation (labels come from the scraping search queries), and therefore we expect many mislabeled examples. Similar to prior work [*e.g.* 12] we train on a subset, **WebVision50**, that contains the first 50 classes ($\approx$ 100,000 images). **Clothing1M** [60] contains clothing images from 14 categories that are also scraped through search queries, as well as a set of "trusted" images annotated by humans. To match the size of WebVision and speed up experiments, we use a 100K subset of the full dataset (**Clothing100K**). For consistency with other datasets, we don't use any trusted images for training or validation. We train ResNet-50 models from scratch on these datasets. We compare against the recent methods of Data Parameters[5], DY-Bootstrap, and INCV.

We use AUM/threshold samples to remove mislabeled training samples and re-train on the cleaned dataset. In Table 3, we compare this approach to a (Standard) model trained on the full dataset. On WebVision50 we flag 17.8% of the data as mislabeled (see Appx. C for examples). Removing these samples reduces error from 21.4% to 19.8%. Similarly, we identify 16.7% mislabeled samples on Clothing100K for a similar error reduction. In comparison, we find that the DY-Bootstrap method tends to estimate less label noise than our method and is unable to reduce error over standard training. DY-Bootstrap mixes the assigned and predicted labels during training; therefore, we hypothesize that it is overconfident with its identifications. INCV tends to remove more data than our method. It achieves higher error on WebVision50, suggesting that it is pruning too aggressively on this dataset.

On the full Clothing1M dataset, AUM reduces error from 33.5% (standard training) to 29.6%. We note that AUM removes fewer data on the full $1M$ dataset than on the $100K$ subset (10.7% versus 16.7%). It is possibly more difficult to identify mislabeled data in larger datasets, as networks require more training to memorize large datasets. We pose this question for future work.

**"Mostly-clean" datasets.** While identification methods should be robust in high-noise settings, they should also work for datasets with few mislabeled examples. To that end, we test our method on *uncorrupted* versions of CIFAR10, CIFAR100, and Tiny ImageNet. While some images might be ambiguous or mislabeled (due to their small size), we expect that most images are correctly-labeled. Additionally, we apply our method to the full ImageNet dataset [13] using ResNet-50 models. In Table 3 we notice several trends. First, using DY-Bootstrap or INCV results in worse performance than a standard training procedure. INCV flags over a quarter of CIFAR100 and Tiny ImageNet as mislabeled, and therefore likely throws away too much data. This suggests that this method is susceptible to *confusing hard (but beneficial) training data for mislabeled data*. DY-Bootstrap tends to remove fewer examples; however, it is again likely that the bootstrap loss is overconfident. Data Parameters provides marginal improvement on ImageNet but has little effect on other datasets.

In contrast, our cleaning procedure *reduces the error* on most datasets. (See Appx. C for examples of removed images.) By removing high AUM data, we reduce the error on CIFAR100 from 33.0% to 31.8%. The amount of removed data differs among the datasets: 3% on CIFAR10, 13% on CIFAR100, and 24% on Tiny ImageNet. Based on these results, we hypothesize that the AUM

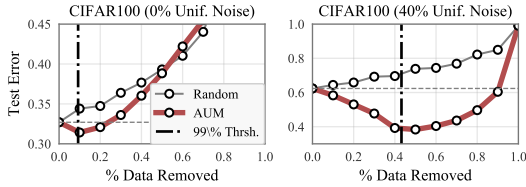

Figure 5: Removing data according to the AUM ranking (red line). A ResNet-32 achieves best test-error when the amount of removed data corresponds to the 99% threshold sample (black line). Removing data randomly results in strictly worse error (gray line).

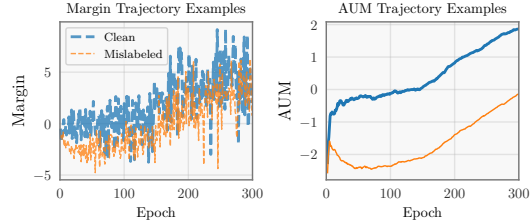

Figure 6: **Left:** Margin trajectories for one clean sample and one mislabeled CIFAR example. The trajectories are difficult to separate consistently. **Right:** AUM trajectories (running average of margin) are separable and less noisy.

method *removes few beneficial training examples, and primarily removes truly-mislabeled data*. On the ImageNet dataset, only 2% of samples are flagged, and removing these samples does not significantly change top-1 error (from 24.2 to 24.4). Given the rigorous annotation process of this dataset, it is not surprising that we find few mislabeled samples.

A list of mislabeled data flagged by AUM is available at `http://bit.ly/aum_mislabeled`.

### 4.4 Analysis and Ablation Studies

**AUM ranking versus margin ranking.** AUM is a running average of a sample's margin. We find that this averaging is necessary for a stable and separable metric. Fig. 6 (left) displays the un-averaged margin for a clean sample and a noisy sample over the course of training (CIFAR10, 40% noise). Note that the two margins occupy a similar range of values and intersect several times throughout training. Conversely, AUM's running average improves the signal-to-noise ratio in the margin trajectories. In Fig. 6 (right), we see a consistent separation after the first few epochs.

**Removing data according to the AUM ranking.** Our method removes data with a lower AUM than threshold samples. Here we study the effect of removing fewer samples (lower threshold), more samples (higher threshold), or random samples. We discard varying amounts of the CIFAR100 dataset and compare models trained on the resulting subsets. We examine 1) discarding data in order of their AUM ranking; and 2) according to a random permutation. Fig. 5 displays test-error as a function of dataset size. Discarding data at random strictly increases test error regardless of threshold. On the other hand, discarding data according to AUM ranking (red line) results in a distinct optimum. This optimum corresponds to the 99% threshold sample (black dotted line), suggesting our proposed method identifies samples harmful to generalization and keeps data required for good performance.

**Effect of data augmentation.** In all the above experiments, we train networks and compute AUM values using standard data augmentation (random image flips and crops). We note that our method is effective even without data augmentation. On CIFAR10 (40% noise) *with* augmentation, AUM reduces error from 43% to 12%; *without* augmentation, it reduces error from 51% to 20%.

**Robustness to architecture and hyperparameter choices.** We find that our method is robust to architecture choice. The AUM ranking achieves 98% Spearman's correlation across networks of various depth and architecture (see Appx. B for details). This suggests that the AUM statistic captures dataset-dependent properties rather than model-dependent properties. Moreover, our method is robust to the choice of threshold sample percentile. For example, if we choose the AUM threshold to be the 90th percentile of threshold samples, the final test-error only differs by 4% (relative)—see Appx. B.

### 4.5 Limitations.

AUM/threshold samples are able to identify mislabeled samples in many real-world datasets. Nevertheless, we can construct challenging synthetic scenarios for our method. One such setting is when mislabelings are extremely systematic: for example, all BIRD images are either correctly labeled or assigned the (incorrect) label DOG (i.e. they are never mislabeled as any other class). To construct such a *asymmetric* noise setting, we assume some ordering of the classes $[1, c]$. With probability $p$, we

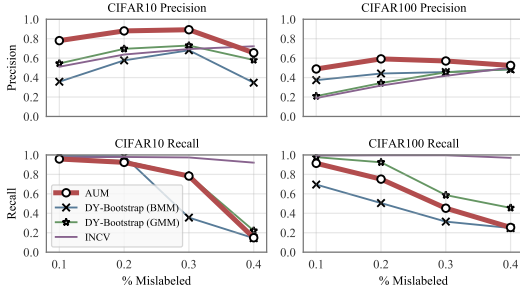

Figure 7: Precision/recall for identifying mislabeled data with asymmetric noise.

Table 4: Test-error (ResNet-32) on datasets with asymmetric noise.

| Dataset | CIFAR10 | | CIFAR100 | |
|---|---|---|---|---|
| Noise | 0.20 | 0.40 | 0.20 | 0.40 |
| Standard | $23.7 \pm 0.1$ | $43.7 \pm 0.0$ | $47.1 \pm 0.2$ | $61.8 \pm 0.0$ |
| Random | $16.1 \pm 0.2$ | $34.8 \pm 1.3$ | $42.3 \pm 0.2$ | $57.1 \pm 0.2$ |
| Bootstrap [50] | $23.8 \pm 0.2$ | $45.0 \pm 0.6$ | $46.6 \pm 0.3$ | $61.3 \pm 0.3$ |
| D2L [40] | $11.4 \pm 0.2$ | $23.6 \pm 1.5$ | $56.4 \pm 0.7$ | $83.1 \pm 1.2$ |
| $L_{DMI}$ [61] | $13.3 \pm 0.8$ | $\mathbf{16.0 \pm 2.1}$ | Diverged | Diverged |
| Data Param [52] | $17.9 \pm 0.2$ | $44.5 \pm 0.7$ | $43.8 \pm 0.5$ | $61.0 \pm 0.4$ |
| DY-Bootstrap [3] | $22.1 \pm 0.1$ | $40.6 \pm 0.6$ | $46.8 \pm 0.0$ | $62.1 \pm 0.0$ |
| INCV [12] | $11.7 \pm 0.1$ | $20.2 \pm 0.4$ | $43.2 \pm 0.1$ | $\mathbf{55.6 \pm 0.7}$ |
| AUM | $\mathbf{10.3 \pm 0.1}$ | $41.3 \pm 0.2$ | $\mathbf{40.3 \pm 0.2}$ | $59.8 \pm 0.1$ |
| Oracle | $9.2 \pm 0.1$ | $10.8 \pm 0.1$ | $35.3 \pm 0.2$ | $38.8 \pm 0.2$ |

alter a sample's assigned label $y$ from its ground-truth class $\widetilde{y}$ to the adjacent class $\widetilde{y} + 1$. In this noise setting, we expect the 99% threshold will be too small. Imagine that BIRDS are mislabeled as DOGS with probability $p = 0.4$. On average, every image with a (ground-truth) BIRD will generate the BIRD prediction with $\approx 60\%$ confidence and DOG with $\approx 40\%$ confidence. Compared to the uniform noise setting, correctly-labeled birds will have a smaller margin and mislabeled birds will have a larger margin. For threshold samples however, the model confidence will be $\approx 1/(c+1)$—representing the frequency of these samples. Threshold sample margins will therefore be smaller than mislabeled margins, resulting in low identification recall.

Our proposed method achieves high precision and recall with 20% asymmetric noise. However, in Fig. 7 we see that recall struggles in the 30% and 40% noise settings. We would note that 40% noise is a nearly maximal amount of noise under this noise model (50% noise would essentially be random guessing), and that the BMM and GMM approaches have a similarly low recall. Table 4 displays test error after discarding data flagged by our method.[6] Though our method outperforms others with 20% noise, it is not competitive with the best method in the 40% setting. A different threshold sample construction—one specifically designed for this noise model—might result in a better AUM threshold that makes our method more competitive. However, given that AUM achieves significant error reductions on real-world datasets (Table 3), we hypothesize that this particular synthetic high-noise setting is not too common in practice.

## 5 Discussion and Conclusion

This paper introduces the AUM statistic and the method of training with threshold samples. Together, these contributions reveal differences in training dynamics that identify noisy labels with high precision and recall. We observe performance improvements in real-world settings—both for relatively-clean and very noisy datasets. Moreover, AUM can be easily combined with other noisy-training methods, such as those using data augmentation or semi-supervised learning. For researchers, we believe that the training phenomena exploited by AUM and threshold samples are an exciting area for developing new methods and rigorous theory. There are several additional directions for future work, such as using the AUM ranking for curriculum learning [e.g. 10, 25, 52], or investigating whether AUM mitigates the double descent phoenomenon for naturally noisy datasets [41].

Importantly, the AUM method works with any classifier "out-of-the-box" without any changes to architecture or training procedure. We provide a simple package (`pip install aum`) that computes AUM for any PyTorch classifier. Running this method simply requires one additional round of model training, which can be easily baked into an existing model selection procedure. Even on relatively clean datasets (like CIFAR100), dataset cleaning with AUM can lead to improvements that are potentially as impactful as a thorough architecture/hyperparameter search. For practitioners, we hope that the "dataset cleaning" step with AUM becomes a regular part of the model development pipeline, as the relatively simple procedure has potential for substantial accuracy improvements.

## Broader Impact

This paper introduces a method to identify mislabeled or harmful training examples. This has implications for two types of datasets. First, it can enable the widespread use of "weakly-labeled" data [e.g. 28, 35, 60], which are often cheap to acquire but have suffered from data quality issues. Secondly, it can be used to audit existing datasets such as ImageNet [13], which are widely used by both researchers and practitioners to benchmark new machine learning methods and applications.

Many research and commercial applications rely on standard datasets for pre-training, like ImageNet [13] or large text corpora [14]. Recent work demonstrates brittle properties of these datasets [9, 49]; therefore, improving their quality could impact numerous downstream tasks. However, it is also worth noting that any automated identification procedure has the potential to create or amplify existing biases in these datasets. Auditing and curation might also have unintended consequences in sensitive applications that require security or data privacy. On common datasets like ImageNet/CIFAR, it is worthwhile to note if identification errors are prone to any particular biases.

## Acknowledgments and Disclosure of Funding

We would like to thank Josh Shapiro for developing our open-source PyTorch package.

## Footnotes

[2] Our package (`pip install aum`) can be used with any PyTorch classification model.

[3] We compare to the DY-Bootstrap variant proposed by [3] that does not use MixUp to disentangle the performance benefits of mislabel identification and data augmentation.

[4] The MentorNet code release can only be used with pre-compiled $0.2/0.4/0.8$ versions of CIFAR.

[5] Note that this method does not explicitly remove data—therefore, we only compare to its final test error.

[6] For baselines, we compare against methods that—like our approach—have no prior knowledge of the noise model. This excludes Co-Teaching, which requires a noise estimate.

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
