[Supplementary Material]

# Supplementary Information for: Identifying Mislabeled Data using the Area Under the Margin Ranking

**Geoff Pleiss**[†]
Columbia University
gmp2162@columbia.edu

**Tianyi Zhang**[†]
Stanford University
tz58@stanford.edu

**Ethan Elenberg**
ASAPP
eelenberg@asapp.com

**Kilian Q. Weinberger**
ASAPP, Cornell University

## A    Experiment Details

All experiments are implemented in PyTorch [48]. Since we don't assume the presence of trusted validation data, we do not perform early stopping. All test errors are recorded on the model from the final epoch of training.

All tables report the mean and standard deviation from 4 trials with different random seeds. On the larger datasets we only perform a single trial (noted by results without confidence intervals).

**CIFAR10, CIFAR100, and Tiny ImageNet.**    All models unless otherwise specified are ResNet-32 models that follow the training procedure of He et al. [21]. We train the models for 300 epochs using $10^{-4}$ weight decay, SGD with Nesterov momentum, a learning rate of 0.1, and a batch size of 256. The learning rate is dropped by a factor of 10 at epochs 150 and 225. We apply standard data augmentation: random horizontal flips and random crops. The ResNet-32 model is designed for $32 \times 32$ images. For Tiny ImageNet (which is $64 \times 64$), we add a stride of 2 to the initial convolution layer.

When computing the AUM to identify mislabeled data, we train these models up until the first learning rate drop (150 epochs). We additionally drop the batch size to 64 to increase the amount of variance in SGD. We find that this variance decreases the amount of memorization, which makes the AUM metric more salient. All other hyperparameters are consistent with the original training scheme.

After removing samples identified by AUM/threshold samples, we modify the batch size so that the network keeps the same number of iterations as with the full dataset. For example, if we remove $25\%$ of the data, we would modify the batch size to be $192$ (down from $256$).

**WebVision50 and Clothing100K.**    We train ResNet-50 models on this dataset from scratch. Almost all training details are consistent with He et al. [21]—$10^{-4}$ weight decay, SGD with Nesterov momentum, initial learning rate of $0.1$, and a batch size of $256$. We apply standard data augmentation: random horizontal flips, random crops, and random scaling. The only difference is the length of training. Because this dataset is smaller than ImageNet, we train the models for 180 epochs. We drop the learning rate at epochs 60 and 120 by a factor of 10.

When computing the AUM, we train models up until the first learning rate drop (60 epochs) with a batch size of 256. As with the smaller datasets, we keep the number of training iterations constant after removing high AUM examples.

---

[†]Work done while at ASAPP.

**Clothing1M and ImageNet.** The ImageNet procedure exactly matches the procedure for WebVision and Clothing1M, except that we only train for 90 epochs, with learning rate drops at 30 and 60. The AUM is computed up until epoch 30.

# B  Additional Ablation Studies

Figure S1: Spearman's correlation of AUM and other metrics across various network architectures (RN=ResNet, DN=DenseNet). AUM (left) produces a very consistent ranking of the training data. The margin itself (middle left) produces less consistent rankings. Training loss (middle right) and validation loss (far right) are also much less correlated across networks.

**Consistency across architectures.** Since AUM functions like a ranking statistic, we compute the Spearman's correlation coefficient between the different networks. In Fig. S1 (far left) we compare CIFAR10 (40% noise) AUM values computed from ResNet and DenseNet [24] models of various depths. We find that the AUM ranking is *essentially the same* across these networks, with $> 98\%$ correlation between all pairs of networks. It is worth noting that AUM achieves this consistency in part because it is a running average across all epochs. Without this running average, the margin of samples only achieves roughly $75\%$ correlation (middle left plot). Finally, AUM is more consistent than other metrics used to identify mislabeled samples. The training loss (middle right plot), used by Arazo et al. [3], achieves $75\%$ correlation. Validation loss (far right plot), used by INCV [12], achieves $40\%$ correlation. These metrics are more susceptible to network variance, which in part explains why AUM achieves higher identification performance.

Figure S2: Test-error of AUM-cleaned models as a function of the threshold sample percentile used to compute the AUM threshold (see Sec. 3).

**Robustness against Threshold Sample Percentile.** As a simple heuristic, we suggest using the 99[th] percentile of threshold sample AUMs to separate clean data from mislabeled data (see Sec. 3). However, we note that AUM performance is robust to this choice of hyperparameter. In Fig. S2 we plot the test-error of AUM-cleaned models that use different threshold sample percentile values. On (unmodified) CIFAR100, we note that the final test error is virtually un-impacted by this hyperparameter. With $40\%$ label noise, higher percentile values typically correspond to better performance. Nevertheless, the difference between high-vs-low percentiles is relatively limited: the $90\%$-percentile test-error is $41\%$, whereas the $99\%$-percentile test-error is $39\%$.

# C   More Results for Real-World Datasets

**AUM values.**   Fig. S3 displays the empirical AUM densities on the real-world datasets. Unlike the synthetic mislabeled datasets (Fig. 3, main text) these datasets do not exhibit bimodal behavior. The 99% threshold sample—represented by a gray line—differs for all datasets.

Figure S3:   AUM distributions on real-world datasets. Gray lines represent the 99% threshold samples.

**Example removed images.**   Fig. S4 displays high AUM images for CIFAR10, CIFAR100, and Tiny ImageNet. Fig. S5 and Fig. S6 display high AUM images for WebVision50, and Clothing1M, respectively.

Figure S4:  Images from CIFAR10 (left) and CIFAR100 (right) with the worst AUM ranking.

Figure S5: Images from WebVision50 with the worst AUM ranking.

Figure S6: Images from Clothing1M with the worst AUM ranking.