[Reviews · NeurIPS 2020]

Review 1

Summary and Contributions: The paper introduced a new metric called AUM--area under the margin. This metric tries to measure the degree of ambiguity or mislabeledness of an example. Consequently, the measure is used to clean up datasets such as CIFAR and Imagenet and to improve performance on respective datasets upon retraining. *** I would like to thank the authors for clarifying my questions. I maintain my support for the paper.

Strengths: The paper achieves improvement over state-of-the-art techniques on an important problem to the community, finding large parts of the dataset that improve performance upon removal. The paper is concise, well-written, and enjoyable to read.

Weaknesses: I found no strong weaknesses in the paper.

Correctness: The claims and methodology seem sound and clean.

Clarity: yes

Relation to Prior Work: Yes, as far as I can tell.

Reproducibility: Yes

Additional Feedback: I would like to thank the authors for a fun paper! I had a few minor questions and remarks. 1) I think there might be an interesting relationship between how clean is a dataset with the recent work on double descent. Double descent is often observed when the data has a natural noise in the distribution or equivalently model misspecification. I conjecture that your method would work proportionally to the peak in test error when looking at epoch-wise double descent. Could be an interesting discussion point in the paper. 2) I'm wondering about the effect of data augmentation on the performance of your method. On one hand, when data augmentation is present, cleaning the dataset from ambiguous examples will have a compounding effect on the performance, as the optimization encounters much more (e.g. flipped) of the bad examples. On the other, data aug serves as regularization, so could be that the model is more smooth in some sense, which will reduce the effectiveness of your method. 3) Given the discussion in "limitations" I would presume that this is not true, but do you think that your method would be able to detect data set poisoning? Minor comments/typos: Line 96: Not sure about the empirical work on margin, but the theoretical work, as far as I know, has very little to do with what actually happens in real models. That is, the margin-based generalization bounds have a prohibitively large norm, rendering the bounds trivial. Figure 2 is slightly truncated Line 266 Why is 2% top 1 error not significant?


Review 2

Summary and Contributions: The paper proposed an area under the margin (AUM) statistic to identify mislabeled samples by measuring the average difference between the logit values for a sample’s assigned class and its highest non-assigned class. Further, the paper proposed adding an extra class populated with especially re-assigned threshold samples to learn the AUM upper bound of mislabeled data. The experimental results on synthetic and real-world datasets show the effectiveness of the proposed method.

Strengths: 1. The AUM method works with any classifier “out-of-the-box” without any changes to architecture or training procedure. 2. The empirical evaluations are sufficient and reproducible.

Weaknesses: 1) The method ignores the classification of the removed data – do the removed samples introduce new problem (e.g., the unbalanced training samples problem)? 2) The method may make unfair comparisons with other methods – in my opinion, the method should be compared with a set of methods doing the same thing, i.e., remove 20% data, and send the remaining to a base learner to check the consequence of removed data. 3) In section 3, the author hypothesizes that “at any given epoch during training, a mislabeled sample will in expectation have a smaller margin that correctly-labeled sample”. Figure 2 uses DOG cases to illustrate this hypothesizes. However, is this the case in general? 4) To find the threshold, a new class is added to estimate the upper bound of AUM for mislabeled data by Eqn (1). However, there are no clean positive samples to learn new classifier, thus the assigned logit value of threshold samples in Eqn (1) will be biased. 5) In Table 4, the results of AUM are worse than that of most compared methods when the degree of noise is 40% - does this indicate that too much data are removed? 6) In practice, how to choose a good set of samples for threshold setting?

Correctness: yes

Clarity: yes, very clear.

Relation to Prior Work: yes

Reproducibility: Yes

Additional Feedback: The authors have addressed my major concerns about leave out datasets. Overall, I think that this paper provides a reasonable and convenient tool for data cleaning. I will keep my score unchanged.


Review 3

Summary and Contributions: The paper addresses the problem of learning with label noise. The authors focus on identifying underlying noisy labels by exploiting the margin ranking. The experimental results on synthetic and real-world datasets verifies the effectiveness of the proposed method. **After reading author response** Thank the authors for their positive response. After reading this response carefully, I felt that part of my confusion had been removed. Thus, I changed my score to 5. However, my main concerns for rejecting remain: Limited contributions and inadequate experimental explanations.

Strengths: This paper uses the margin ranking to identify noisy labels, which prevents the memorzation of label noise. The idea is simple and effective. The experimental settings are detailed, which is easy to follow. The results of the synthenic experiments are enough and convincing.

Weaknesses: 1. Limited novelty. The authors use the margin ranking to find the data with noisy labels. As far as know, the concept of margin has long been used for classification task, e.g., face recognition [1], [2], semi-supervised learning [3]. The methods [4], which employ memorization effect to select confident samples (small loss samples), also share similar ideas. The data with small loss has clean labels with high confidence and also has a larger margin ranking. The authors ignore the discussion and comparison about these existing works. 2. The logic and contribution are not clear enough. The proposed method seems simple, but the descriptions make readers confused. Thus, I would like to see an algorithm flow and suggest the authors can re-emphasize the contributions of the paper. 3. Missing some necessary analysis. For example, the authors use a 100,000 sample subset of Clothing1M. They state the goal is to match the size of Webvision dataset. The training data in the different datasets is usually distributed differently and has different challenge to the proposed method. Maybe the authors can add some discussion to these settings. Besides, I would like to know the detailed analyses about the difference AUM and original margin, which is critical to readers. [1] Hao Wang et al. CosFace: Large Margin Cosine Loss for Deep Face Recognition. CVPR 2018. [2] Jiankang Deng et al. ArcFace: Additive Angular Margin Loss for Deep Face Recognition. arxiv 1801.09414. [3] Jinhao Dong et al. MarginGAN: Adversarial Training in Semi-Supervised Learning. NeurIPS 2019. [4] Bo Han et al. Co-teaching: Robust training of deep neural networks with extremely noisy labels. NeurIPS 2018.

Correctness: Yes, the claims and method are correct.

Clarity: No. As for me, the paper is not well written. Some issues have been mentioned in "Weaknesses".

Relation to Prior Work: No. It is not clearly discussed, which is also metioned in "Weaknesses".

Reproducibility: Yes

Additional Feedback:


Review 4

Summary and Contributions: This paper focuses on label noise learning problems in weakly-supervised learning. The authors propose a novel data cleansing method based on the AUM statistic, which can automatically distinguish mislabeled sample from the correct ones.

Strengths: 1) This paper is self-contained and well-structured. For example, based on previous works and examples, the authors introduce the dynamics of SGD in distinguishing correct/incorrect signals. In Sec 3, the authors verified these assumptions via AUM, and proposed the corresponding data cleansing algorithm. 2) Using threshold sample in determining hyperparameters might be a good idea, as it can save a lot of computational demands. Fig 3 and Fig 5 verified this strategy on synthetic datasets with label noise.

Weaknesses: 1) The authors claimed that the AUM is less prone to confusing difficult samples for mislabeled data (P2-68), I think it is a good idea to provide some explanations or heuristics in comparison with previous works. Moreover, data cleansing has been widely exploited in the literature of label noise learning. So, maybe it is better to demonstrate the superiority of the proposed AUM-based methods over the counterparts. 2) The experimental results on Clothing1M is not very convincing. Using 10% of the training data, the test accu of sota methods (e.g., PENCIL Kun Yi et al. ) can reach 73% with backbone ResNet-50. However, the test accu of AUM is only 67%. Could the authors provide some explanation? Maybe the sampled data in the experiment are imbalanced. 3) The authors claimed that non-uniform label noise is not too common in practice. However, in real-world noise datasets, such as Clothing1M, it has been verified that non-uniform label noise widely exists. Kun Yi, Jianxin Wu; Probabilistic End-To-End Noise Correction for Learning With Noisy Labels. Proceedings of the IEEE/CVF Conference on Computer Vision and Pattern Recognition (CVPR), 2019, pp. 7017-7025

Correctness: correct

Clarity: well written

Relation to Prior Work: clear

Reproducibility: Yes

Additional Feedback:

[Author Response · NeurIPS 2020]

We thank the reviewers for their helpful feedback. We are encouraged that you note AUM's simplicity—"works with any classifier out-of-the-box" (R2), "it can save a lot of computational demands" (R4)—as we believe this differentiates our method from existing ones. We would also emphasize our results on supposedly-clean real-world datasets; for example, a $1.2\%$ reduction in error on CIFAR100 (without synthetic noise) simply by removing data.

**It seems that R3, as they admit themself, is "confused" by our submission and contribution.** *Clearly our use of the label margin concept is not intended to be a novelty in and of itself, as it has been prominent in the ML literature for decades with uses in hundreds of publications.* We could cite [Wang et al., CVPR 2018] (as suggested by R3) but we find the earlier work more appropriate, like [Vapnik, 1995], [Bartlett, NeurIPS 1997], or [Weinberger and Saul, JMLR 2009]. The novelty of our method is using the label margin as part of an intuitive and reliable metric to identify mislabeled data, which we are the first to do. Additionally, we clearly discuss/compare to Co-Teaching in Sec. 2/Table 1, and note that the other reviewers find our paper "well written" (R4), "enjoyable to read" (R1), and "very clear" (R2). However, we do agree with R3's point concerning the subsampled Clothing1M dataset (see response to R4).

**R1.** Thank you for your supportive comments and interesting remarks. Per your suggestions, we will discuss the connection between double descent and detecting dataset poisoning in Sec. 5. (*"Effect of data augmentation."*) AUM performs comparably with/without augmentation. On CIFAR10 ($40\%$ noise) *with* augmentation, AUM reduces error from $43\%$ to $12\%$; *without* augmentation, it reduces error from $51\%$ to $20\%$. (*"Line 266 Why is $2\%$ top 1 error not significant?"*) This is a typo; thank you for catching. On ImageNet the standard network achieves a top-1 error of $24.2\%$ (as in Table 3, not $22.2\%$ as in line 226). Thus the difference between AUM and standard training is $0.2\%$.

**R2.** Thank you for positive feedback and detailed questions. We hope to address them here and in the camera ready.

*Fair comparisons with other methods*: Our results in Fig. 3 indicate that AUM identifies mislabeled samples with higher precision and recall than other methods. Therefore, if we use our training procedure (remove identified data, send remaining data to a base learner) AUM should outperform existing identification methods.

*"Do the removed samples introduce new problem?"*: This is an interesting point. Empirically, some classes in e.g. WebVision are less likely to be mislabeled (e.g. GOLDFISH) than others (e.g. WATER OUZEL). The number of samples removed per class varied from $109$ to $828$. We note that this dataset was already imbalanced (class size ranging from $701$ to $5688$); therefore it is unlikely AUM introduced a new imbalance problem. We will discuss this more in Sec. 5.

*"How to choose a good set of [threshold] samples?"*: We choose $N/(C+1)$ threshold samples (for $C$ classes) uniformly at random from the training set (see line 135). This way, the threshold class size equals the average class size, though this strategy might need adjustment for extremely imbalanced datasets. We are unclear what you mean by "the assigned logit value of threshold samples will be biased." Threshold samples approximate mislabeled samples, as a large threshold logit cannot be learned through generalization (i.e. no "true" positives exist) and therefore must be memorized.

**R3.** *"Analyses about the difference AUM and original margin"*: AUM is more robust and consistent than the margin at any given epoch. Averaging across epochs increases the "signal to noise ratio." See Fig. S1, S3 in the appendix.

**R4.** Thank you for pointing us to Yi and Wu's PENCIL paper! Although quite different from AUM, it is clearly relevant in this context and we will of course include it in the camera ready version.

*"The authors claimed that non-uniform label noise is not too common in practice"*: Sorry, this is not what we meant in line 313 ("this particular high-noise setting is not too common"). We were referring to $40\%$ pair-wise asymmetric noise, which is an extreme and synthetic setting. We completely agree that non-uniform noise does exist, and our method is able to successfully reduce error on real-world (non-uniform) noisy datasets (Table 3).

*Supporting the claim "AUM is less prone to confusing difficult samples for mislabeled data"*: Table 3 provides evidence for this claim, though we agree the text in Sec. 4 should emphasize this more. On CIFAR10/100/ImageNet (without any synthetic noise, Table 3), INCV and DY-Bootstrap achieve worse performance than standard training, suggesting that some of the large-loss samples removed by these methods were actually "good" data. Our method actually improves accuracy over standard training, suggesting that AUM is not removing these difficult (but beneficial) data.

*Clothing1M results*: On the full Clothing1M dataset, AUM achieves $29.0\%$ error (standard training achieves $31.1\%$). Originally we trained on a 100K subset due to a limited computational budget; we will update Table 3 with the full dataset results. While AUM does not achieve SOTA on this task (compared to e.g. PENCIL), we emphasize that it consistently improves error both on noisy datasets (WebVision, Clothing1M) and clean datasets (CIFAR, TinyImageNet).

*"Data cleansing has been widely exploited in the literature of label noise."* We are not sure which "data cleansing" methods you are referring to, but please let us know which additional baselines we should include in the final version. Currently, we compare AUM's identification performance against INCV and DY-Boostrap (Fig. 3 and 6).

[Meta-Review · NeurIPS 2020]

This paper proposes a simple strategy for identifying training samples that may be mislabeled. The paper received mixed reviews from the reviewers, with one reviewer in particular strongly arguing that simple and effective methods are deserving of publication, while other reviewers were concerned that the approach is too straightforward. Weighing these arguments, it was felt in discussion that this paper could be of interest as a straightforward approach to deal with the important problem of label noise.